# Feature Construction Using Persistence Landscapes for Clustering Noisy IoT Time Series

**Renjie Chen** [†] **and Nalini Ravishanker** [*,†]

Department of Statistics, University of Connecticut, Storrs, CT 06269, USA; renjie.chen@uconn.edu
* Correspondence: nalini.ravishanker@uconn.edu
† These authors contributed equally to this work.

**Abstract:** With the advancement of IoT technologies, there is a large amount of data available from wireless sensor networks (WSN), particularly for studying climate change. Clustering long and noisy time series has become an important research area for analyzing this data. This paper proposes a feature-based clustering approach using topological data analysis, which is a set of methods for finding topological structure in data. Persistence diagrams and landscapes are popular topological summaries that can be used to cluster time series. This paper presents a framework for selecting an optimal number of persistence landscapes, and using them as features in an unsupervised learning algorithm. This approach reduces computational cost while maintaining accuracy. The clustering approach was demonstrated to be accurate on simulated data, based on only four, three, and three features, respectively, selected in Scenarios 1–3. On real data, consisting of multiple long temperature streams from various US locations, our optimal feature selection method achieved approximately a 13 times speed-up in computing.

**Keywords:** elbow method; feature construction; IoT time series; persistence landscape; topological data analysis; unsupervised learning

## 1. Introduction

Enhanced IoT technologies have been developing at a remarkable pace, allowing long streams of data to be collected from a large number of in situ wireless sensor networks. Application domains include business, biomedicine, energy, finance, insurance, and transportation sensors being installed across broad geographic regions. Data streams collected by these sensors constitute long, noisy time series with complex temporal dependence patterns, leading to several different types of interesting and useful data analysis. For example, specially adapted machine learning techniques for anomaly detection in internal temperatures, by sensors placed inside thousands of buildings in the US by an insurance company, were developed in [1], with the goal of alerting clients, and attempting to mitigate the risk of pipe freeze hazard. An extended analysis of the same IoT streams was provided in [2], by employing a Gaussian process model framework, to assess the causal impact of a client reaction to an alert. Data analysis of energy usage, management, and monitoring on a large academic campus was described in [3]. The role of wireless sensor technologies in the agriculture and food industry was discussed in [4].

There is also considerable interest in analyzing IoT streams to understand different aspects of weather monitoring and climate change. For example, remote sensing in water environmental processes was discussed in [5], while [6] discussed how inexpensive open-source hardware is democratizing (climate) science, because open-source sensors are able to measure environmental parameters at a fraction of the cost of commercial equipment, thus offering opportunities for scientists in developed and developing countries to analyze climate change at both local and global regional levels. A report from the United Nations Intergovernmental Panel on Climate Change (IPCC) states that average temperatures

are likely to continue rising, even with mitigating efforts in place (https://www.ipcc.ch/report/ar6/wg1/ (accessed on 1 October 2019)). NOAA observation systems collect data twice every day, from nearly 100 locations in the US. The National Weather Service (NWS) launches weather balloons, carrying instrument packages called radiosondes. Radiosonde sensors measure upper air conditions, such as atmospheric pressure, temperature and humidity, and wind speed and direction. The Automated Surface Observing Systems (ASOS) program is a joint effort by the National Weather Service (NWS), the Federal Aviation Administration (FAA), and the Department of Defense (DOD). The ASOS system serves as the primary surface weather observing network in the US, updating observations every minute (https://www.weather.gov/about/ (accessed on 1 October 2019)) (https://nosc.noaa.gov/OSC/ (accessed on 1 October 2019)).

When weather data are available from a large number of locations, clustering/grouping locations based on stochastic properties of the data are of considerable interest [7–10]. To this end, it is useful to develop effective algorithms that construct useful features that capture the behavior of the time series: clustering then proceeds on the basis of similarity/dissimilarity metrics between the features. There is a considerable literature on feature-based time series clustering. For example [11], categorized feature representations for time series fall into into four broad types: (i) data-adaptive representations, which are useful for time series of arbitrary lengths; (ii) non-data-adaptive approaches, which are used for time series of fixed lengths; (iii) model-based methods, which are used for representing time series in a stochastic modeling framework; and (iv) data-dictated approaches, which are automatically defined, based on raw time series. In this article, we describe feature construction based on persistent homology, a concept in topological data analysis (TDA), which we used to cluster locations with similar weather patterns.

Topological data analysis [12] encompasses methods for discovering interesting shape-based patterns, by combining tools from algebraic topology, computer science, and statistics, and it is becoming an increasingly useful area in many time series applications. For a review of persistent homology for time series, and a tutorial using the R software, see [13]. In particular, the review discusses ideas such as transforming time series into point clouds via Takens embedding [14], creating persistence diagrams [15,16], and constructing persistence landscapes of all orders [17]. While persistent homology is a central tool in TDA, for summarizing geometric and topological information in data using a persistence diagram (or a bar code), it is cumbersome to construct useful statistical quantities using metrics such as the Wasserstein distance. Persistence landscapes enable us to map persistent diagrams to a Hilbert space, thereby making it easier to apply tools from statistics and machine learning. Recent research has explored the use of persistence landscapes as features to either cluster or classify time series [18–20]. Persistence landscapes of *all orders* using weighted Fourier transforms of continuous-valued EEG time series were constructed by [21], and used as features for clustering the time series, using randomness testing to examine the robustness of the approach to topology-preserving transformations, while being sensitive to topology-destroying transformations. *First-order* persistence landscapes, constructed from Walsh–Fourier transforms of categorical time series from a large activity–travel transportation data set, were used by [22] to create features for clustering, arguing that the first-order landscape was sufficient for accurately clustering time series with relatively simple dependence properties. Several aspects of using persistence homology in time series analysis have been discussed in-depth in [23].

It is well-known that lower-order persistence landscapes contain more important topological features than higher-order landscapes, which are closer to zero [17]. In many situations, it may be unnecessary and computationally prohibitive to use persistence landscapes of all orders to elicit useful features of time series. Selecting the order of the persistence landscapes to serve as features requires a delicate balance between missing important signals and introducing too much noise. Existing research has not addressed the problem of data-based selection of the *order* of persistence landscapes that is sufficient to yield accurate clustering: thus, the focus of this article was to address the problem of

deciding the order of persistence landscapes in a time series clustering scenario, and to study this question in the context of noisy, periodic stationary time series, using the smoothed second-order spectrum to construct persistence landscapes. The solution was an algorithm which automatically selected the optimal order of persistence landscapes in a sequential way. These features were then used in clustering the time series. The computational gain from the algorithm was demonstrated through extensive simulation studies, which showed a speed-up of approximately 13 times. We then illustrate our approach, using long temperature streams from from different US locations, and show that features constructed from the selected orders of persistence landscapes produced meaningful clusters of the locations, which may be useful for climate scientists in a comparative study of temperature patterns over time in several locations.

The format of the paper is as follows. Section 2 provides the background of persistence homology for time series, and specifically describes the construction of persistent landscapes. Section 3 discusses literature that is closely related to our research. Section 4 then presents the motivation for our research, and highlights our contribution. Section 5 describes the construction and implementation of our proposed algorithm for feature construction, which is used for time series clustering. Section 6 presents a simulation study, to evaluate the performance of our clustering method. Section 7 clusters daily maximum temperatures across various locations in the United States. Section 8 summarizes the paper.

## 2. Background

In this section, we briefly present the background of persistence homology for time series, which is relevant to our objective of clustering time series based on their stochastic properties.

### 2.1. Time Series to Second-Order Spectrum

Let $x_t$, $t = 1, \ldots, T$ be $T$ observations from a stationary time series $\{X_t\}$ with second-order spectral density function $f(\omega)$. Let $\iota = \sqrt{-1}$. The tapered discrete Fourier transform (DFT) and the corresponding second-order periodogram of $X_t$ are defined as [24]

$$d(\omega_j) = T^{-1/2} \sum_{t=1}^{T} h_t x_t e^{-2\pi\iota\omega_j t}, \text{ and} \tag{1}$$

$$I(\omega_j) = |d(\omega_j)|^2, \tag{2}$$

where $\omega_j = j/T$, $j = 1, 2, \ldots, T/2$ are the Fourier frequencies. It is well-known that $I(\omega_j)$ is not a consistent estimate of the true spectrum $f(\omega)$ of the time series $X_t$: a remedy is to smooth the periodogram $I(\omega)$, in order to get a consistent estimate of the true spectrum. Several smoothing kernels can be employed, including the Daniell, modified Daniell, Bartlett, and Hanning kernels. Let $\tilde{I}(\omega)$ be the smoothed periodogram, using a modified Daniell kernel over $2m + 1$ points surrounding $\omega_j$ as

$$\tilde{I}(\omega) = \sum_{k=-m}^{m} c_k I(\omega_j + k/T),$$

where the weights $c_k$ at the two endpoints receive half the weight that the interior points receive; see [24–26] for more details.

### 2.2. Spectrum to Persistence Diagram

Starting from the smoothed spectrum $\tilde{I}(\omega_j)$, the R function `gridDiag()` in the `R-TDA` package computes the persistence diagram, which shows the births and deaths of the connected components of the time series $X_t$, denoted by $\tilde{\Omega} = \{(\lambda_{k,1}, \lambda_{k,2}), k = 1, 2, \ldots, \tilde{K}\}$ [13]. The function `gridDiag()` is useful for computing the persistent homology of sublevel sets (or superlevel sets) of functions evaluated over a grid of points; specifically: (a) it computes the periodogram function in Equation (2) over a triangulated grid; (b) it constructs a fil-

tration of simplices, using the values of the function; and (c) it computes the persistent diagram. (https://cran.r-project.org/web/packages/TDA/vignettes/article.pdf (accessed on 1 June 2019))

*2.3. Persistence Diagram to Persistence Landscape*

Next, we construct persistence landscapes from the persistence diagrams, using the R function `landscape()` in the `R-TDA` package. Persistence landscapes provide useful summaries of topological properties, and are easy to combine with tools from statistics and machine learning. It was proved by [17] that a set of persistence landscapes admits a unique mean, and preserves the statistical stability of the data distribution, and that using persistence landscapes can preserve differences in the persistence diagrams. The function `landscape()` takes as inputs a persistence diagram $\tilde{\Omega}$ (`Diag`), its dimension $p$ (`dimension`), the landscape order $\nu$ (`KK`) with default value 1, the value of $L$ (`length`), and the region of $L$ (`tseq`); it outputs the persistence landscape of each order $\nu$ as a vector $PL_\nu(\ell)$, $\ell = 1, \ldots, L$. If we set $\nu = 40$, for example, the function will produce a persistence landscape for each $\nu$. See [12,17,27] for more details, and [13] for the R code for constructing persistence diagrams and persistence landscapes from estimated spectra.

## 3. Related Work

Clustering locations or climate stations based on temperature or precipitation time series have been discussed in several recent articles. A two-step cluster analysis of 449 southeastern climate stations was described in [7], to determine general climate clusters for eight southeastern states in the US, and has been employed in several follow-up analyses involving the classification of synoptic climate types. In a similar vein, ref. [8] used a hierarchical cluster analysis to demarcate climate zones in the US, based on weather variables, such as temperature and precipitation.

Spatial grouping of over 1000 climate stations in the US was discussed in [10], by using a hybrid clustering approach, based on a measure of rank correlation as a metric of statistical similarity. Based on the clustering temperatures at these stations, they showed that roughly 25% of the sites accounted for nearly 80% of the spatial variability in seasonal temperatures across the country.

Recent research has used persistence landscapes constructed from time series as features to cluster the time series. In this section, we review related works that focus on using persistent homology for clustering time series. We then highlight the novelty and usefulness of our approach.

A framework for implementing the break detection of critical transitions on daily price cryptocurrencies, using topological data analysis (TDA), was proposed by [19]. They (i) transformed the time series into point clouds, using Taken's delay embedding, (ii) computed persistence landscapes for each point cloud window, (iii) converted to their $L_1$ norms, and (iv) used K-means clustering for these windowed time series.

Persistence landscapes of all orders were employed by [20] as topological features for time series, via Taken's time-delayed embedding transformation, and principle component analysis for denoising the time series.

The problem of clustering continuous-valued EEG time series was studied by [21]. They constructed weighted Fourier transforms of the time series, and constructed persistence landscapes of all orders: they used these as features for clustering the time series. They also examined the robustness of their approach to topology-preserving transformations, while being sensitive to topology-destroying transformations.

The use of persistence landscapes for clustering categorical time series from a large activity–travel transportation data set was described in [22]. They first constructed Walsh–Fourier transforms of the categorical time series, and then obtained first-order persistence landscapes from the Walsh–Fourier transforms, which they used as features for clustering the time series. In this case, they argued that the first-order landscape was sufficient for

accurately clustering time series with relatively simple dependence properties, as in the activity–travel transportation data.

An analysis of multivariate time series using topological data analysis was proposed in [28], by converting the time series into point cloud data, calculating Wasserstein distances between the persistence diagrams, and using the *k*-nearest neighbors algorithm for supervised machine learning, with an application to predicting room occupancy during a time window.

Time series clustering with topological–geometric mixed distance (TGMD) was discussed in [29], which jointly considered the local geometric features and global topological characteristics of time series data. The results revealed that their proposed mixed-distance-based similarity measure could lead to promising results, and to better performance than standard time series analysis techniques that consider only topological or geometrical similarity.

In our approach, we constructed persistent landscapes from the second-order spectra of stationary time series, with the goal of clustering a large number of time series. To balance computing time and clustering accuracy, we developed an algorithm to select an optimal number of landscapes to be employed as features in the clustering.

## 4. Motivation and Contributions

In the previous sections, we discussed issues regarding employing features based on persistence landscapes for clustering time series. Persistence landscapes constitute representations of topological features in a Hilbert space: hence, using landscapes of all orders as features may be the most informative approach in statistical learning [17]; however, constructing landscapes of all orders may be computationally expensive. In general, lower-order persistence landscapes contain information about important topological features of the data, while higher-order persistence landscapes are closer to zero, and generally represent noise.

If $\tilde{K}$ denotes the number of local minima in the smoothed periodogram $\tilde{I}(\omega_j)$, then in computing its persistence landscapes $PL_v(1), \ldots, PL_v(L)$ of all orders $v$, a large fraction of the computing time is used for sorting the $\tilde{K}$ topologies for each value of $\ell$. When the time series $X_t$ has complex signals (perhaps consisting of a complicated mixture of several patterns), or $X_t$ is very noisy (perhaps with a small signal-to-noise ratio, and/or time-varying variances), then $\tilde{K}$ is likely to be large, and it may be important to construct landscapes of large order, to pull out information useful for learning the time series. However, for time series with simpler stochastic patterns, the order of landscapes need not be very high, and computing them may be costly, but wasted effort. In summary, while the use of persistence landscapes for time series clustering is a useful and novel idea, the method can be time-consuming if landscapes of all orders are used, and inaccurate if too few (say, one) landscapes are used as features.

In particular, when we cluster or classify a large number $N$ of time series, an algorithm to decide the order of useful persistence landscapes is useful. There is no approach in the existing literature, to our knowledge, for determining the optimal number of persistence landscapes as features for clustering time series: this motivated our work to explore this question and contribute to the literature in this regard.

Specifically, the contribution of this research is the development of an algorithm to select the smallest order of persistence landscapes to be employed as features for clustering a large number of time series, in order to speed up the computing without sacrificing accuracy. The implementation is described in the next section.

## 5. Implementation

This section describes optimal feature construction from persistence landscapes corresponding to time series. Specifically, our research describes an algorithm for selecting the *smallest order of persistence landscapes* to be employed as features for clustering/classifying a large number of time series, in order to speed up the computing without sacrificing

accuracy. Depending on the nature of the time series, the selected number could be one or a small integer—say, two or three. Our algorithm takes advantage of the monotonicity and non-negativity of persistence landscapes, to set up a scoring function that accumulates over the different orders. This is similar to the behavior of the total within-cluster sum of squares, as the number of clusters increases in unsupervised learning, where the *elbow method* is used to select the appropriate number of clusters [30].

For $n = 1, \ldots, N$, let $\tilde{I}^{(n)}(\omega_j), j = 1, \ldots, T/2$ be the smoothed spectrum, $\tilde{\Omega}^{(n)} = \{(\lambda_{k,1}^{(n)}, \lambda_{k,2}^{(n)}), k = 1, 2, \ldots, \tilde{K}\}$ be the persistence diagram, and $\mathrm{PL}_\nu^{(n)}(\ell), \ell = 1, \ldots, L$ be the $\nu$th order persistence landscapes. Algorithm 1 describes an approach to selecting the optimal (smallest) order $\nu = \nu^{\mathrm{opt}}$, in the context of clustering $N$ time series, each of length $T$: specifically, it chooses the farthest point $(\nu^{\mathrm{opt}}, S_{\nu^{\mathrm{opt}}})$ from the function formed by using points $(1, S_1)$ and $(\nu, S_\nu)$.

The features $\mathrm{PL}_\nu^{(n)}(\ell), \nu = 1, 2, \ldots \nu^{\mathrm{opt}}; \ell = M_1, \ldots, M_2$ obtained from Algorithm 1 for each time series represent the important topological features of the data. Using these features within a K-means algorithm produces effective clustering of the $N$ time series in less computing time than if all the persistence landscapes were employed as features. Let

$$D(n_1, n_2) = \sqrt{\sum_{\nu=1}^{\nu^{\mathrm{opt}}} \sum_{\ell=M_1}^{M_2} (PL_\nu^{(n_1)}(\ell) - PL_\nu^{(n_2)}(\ell))^2}$$

denote the Euclidean distance between the persistence landscapes of the $n_1$-th and $n_2$-th time series. Smaller (larger) Euclidean distance implies that the two series have similar (dissimilar) topological features.

---

**Algorithm 1** Feature Construction Using Persistence Landscapes

---

**Input:** Set of time series $\{X_t^{(n)}, t = 1, \ldots, T, n = 1, 2, \ldots, N\}$; $N$ is the number of time series, each of length $T$.
**for** $n = 1$ **to** $N$ **do**
   Compute $\tilde{I}^{(n)}(\omega_j), j = 1, \ldots, T/2$ by using the R function `spec.pgram()` with a modified Daniell kernel.
   Compute the persistence diagram $\tilde{\Omega}^{(n)} = \{(\lambda_{k,1}^{(n)}, \lambda_{k,2}^{(n)}), k = 1, 2, \ldots, \tilde{K}\}$ from $I^{(n)}(\omega_j)$ by using the R function `gridDiag()` in *R-TDA* package.
**end for**
Compute $M_1 = \min_{n,k} \lambda_{k,1}^{(n)}$ and $M_2 = \max_{n,k} \lambda_{k,2}^{(n)}$.
Initialize PL order $\nu = 1$,   flag = **true**, $\nu^{\mathrm{opt}} = 1$ when $\nu = 1$.
**while** flag **do**
   **for** $n = 1$ **to** $N$ **do**
      Compute the $\nu$-th order persistence landscapes $\mathrm{PL}_\nu^{(n)}(\ell), \ell = 1, \ldots, L$.
   **end for**
   Compute $S_\nu = \sum_{\ell=M_1}^{M_2} \sum_{n=1}^{N} \mathrm{PL}_\nu^{(n)}(\ell)$.
   **if** $\nu > 1$ **then**
      Fit linear function $y = ax + b$ of $(x, y)$ on points $(1, S_1)$ and $(\nu, S_\nu)$, so that $\hat{a} = \frac{S_\nu - S_1}{\nu - 1}, \hat{b} = S_1 - \hat{a}$.
      **for** $\nu^+ = 1$ **to** $\nu$ **do**
         Compute $D(\nu^+) = \frac{|\hat{a}\nu^+ + \hat{b} - S_{\nu^+}|}{\sqrt{\hat{a}^2 + 1}}$ as Euclidean distances of points $(\nu^+, S_{\nu^+})$ to the fitted linear function above.
      **end for**
      Calculate $\nu^{\mathrm{opt}} = \arg\max_{\nu^+ \leq \nu} D(\nu^+)$ (let $\nu^{\mathrm{opt}} = 1$ when $\nu = 2$ since $D(1) = D(2) = 0$ when $\nu = 2$).
      **if** $\nu \geq 3$ **and** $\nu^{\mathrm{opt}} = (\nu - 1)^{\mathrm{opt}} = (\nu - 2)^{\mathrm{opt}}$ **then**
         flag = **false**
      **end if**
   **end if**
   $\nu = \nu + 1$
**end while**
**Output:** optimum order $\nu^{\mathrm{opt}}$, and $\mathrm{PL}_{\nu_0}^{(n)}(\ell), \nu_0 = 1, 2, \ldots \nu^{\mathrm{opt}}; \ell = M_1, \ldots, M_2$ as the feature representation for the $n$-th time series.

---

We implement K-means clustering via the function `kmeans()` in R, using multiple random initializations with `nstart=10` [31,32]. Section 6 implements the landscape order selection algorithm and K-means clustering for $N$ simulated time series, while Section 7 illustrates the approach on daily temperatures from monitoring sensors.

## 6. Simulation Study

We simulated time series of length $T = 1024$ generated from $J$ different populations. Each time series $X_{j,t}^{(n)}$ within the $j$-th population contained a (population specific) signal $Y_{j,t}$, which was a mixture of $M_p$ periodic components together with additive white noise $\epsilon_{j,t}^{(n)}$. The scenarios varied according to their signal-to-noise ratio, defined by $\mathrm{SNR} = \mathrm{Var}(Y_{j,t})/\mathrm{Var}(\epsilon_{j,t}^{(n)})$. The set of $M_p$ signal frequencies and amplitudes differed between the $J$ populations. We studied single-frequency signals and signals that were a mixture of three different frequencies. Furthermore, within each population, we slightly perturbed the signal frequencies around the selected fixed values, in order to simulate realistic situations: as a result, each of the $n$ time series within the same population were allowed to have nearly similar, though not identical, signals.

Step 1. Let the number of populations be $J = 2$. We selected three different values of SNR, i.e., 0.5, 1.5, and 3.0. Corresponding to each SNR value, we generated periodic time series, with frequencies and amplitudes described below:

(a) Scenario 1 with $M_p = 3$. We selected three periods in the first population, to be $(180, 90, 14)$ (i.e., half-year cycle, three-month cycle, and two-week cycle), and in the second population, as $(120, 30, 7)$ (i.e., four-month cycle, one-month cycle, and one-week cycle). We selected the amplitudes at these frequencies, to be $A_{j,u} = (2, 1, 0.5)$ in the first population, and $(1.5, 1.5, 0.5)$ in the second population, for $u = 1, \dots, M_p$;

(b) Scenario 2 with $M_p = 3$. We selected three periods in the first population, to be $(30, 14, 7)$ (i.e., one-month cycle, two-week cycle, and one-week cycle), and in the second population, as $(120, 30, 14)$ (i.e., four-month cycle, one-month cycle, and two-week cycle). We select the amplitudes at these frequencies, to be $A_{j,u} = (2, 1, 0.5)$ in the first population, and $(0.1, 0.15, 2.5)$ in the second population, for $u = 1, \dots, M_p$;

(c) Scenario 3 with $M_p = 1$. We selected the period, as 180 in the first population, and 120 in the second population. We selected the amplitude in the first population, as $A_{1,1} = 2$, and the amplitude in the second population, as $A_{1,1} = 1.5$.

With these parameters, we generated $Y_{j,t} = \sum_{u=1}^{M_p} A_{j,u} \cos(2\pi t / f_{j,u})$, to represent the simulated signal in the $j$-th population.

Step 2. We simulated $N_j = 20$ time series in each of $J = 2$ populations. The period was not fixed, but randomly varied about the fixed periods selected for each population:

(a) We simulated $M_p$ random periods that uniformly varied around the fixed periods selected for each population, i.e., $\delta f_{j,u} \sim \mathtt{Uniform}(a_{j,u}, b_{j,u})$, where we set $a_{j,u} = 0.8 f_{j,u}$ and $b_{j,u} = 1.2 f_{j,u}$;

(b) We simulated a time series in the $j$-th population as

$$X_{j,t}^{(n)} = \sum_{u=1}^{M_p} A_{j,u} \cos(2\pi t / \delta f_{j,u}) + \epsilon_{j,n,t}, \text{ where } \epsilon_{jt}^{(n)} \sim N\left(0, \frac{\mathrm{Var}(Y_{j,t})}{\mathrm{SNR}}\right),$$

which generated $N_j = 20$ time series in each of the $J = 2$ populations.

Step 3. We constructed features based on persistence landscapes in the $N = N_1 + N_2$ series, as follows:

(a) based on $\nu = 1$ only;

(b)  based on $\nu = 1, \ldots, \nu^{\text{opt}}$, using Algorithm 1;
(c)  based on $\nu = 1, \ldots, 40$, to reflect *all* orders.

Step 4.  In each of the cases above, we used K-means clustering, using the `kmeans()` function in R with $C = 2$ clusters, and we computed the accuracy of clustering, via

$$A_{\text{SNR}} = \frac{\#\{\texttt{correct labels}\}}{\sum_{j=1}^{J} N_j},$$

where the numerator denoted the number of correct labels (classification).

Step 5.  We repeated Steps 2–4 for a total of $R = 50$ times, and computed the average accuracy across the replications, as $\overline{A}_{\text{SNR}}$.

The average classification accuracy corresponding to each SNR value in each of the three scenarios, and each feature construction approach (smoothed tapered spectrum, PL(1), PL($\nu$), $\nu = 1, \ldots, \nu^{\text{opt}}$, and PL($\nu$), $\nu = 1, \ldots, 40$) is shown in Table 1. Note that there is no upper bound on the PL order $\nu$, and that we chose $\nu = 40$ as a sufficiently large number. In Scenario 1, when $M_p = 3$, the elbow method chose $\nu^{\text{opt}} = 4$, and the features were constructed using persistence landscapes of orders $\nu = 1, \ldots, 4$; whereas, in Scenario 2, $\nu^{\text{opt}} = 3$ was selected. When $M_p = 1$, $\nu^{\text{opt}} = 3$ was selected. The use of $\nu^{\text{opt}}$ was able to preserve almost all the information that was preserved when using all orders, and performed best in most cases.

**Table 1.** Comparisons of the average accuracy of clustering. Different PL orders corresponding to three SNR Values.

| Scenarios | Scenario 1 ($\nu^{\text{opt}} = 4$) | | | Scenario 2 ($\nu^{\text{opt}} = 3$) | | | $M_p = 1$ ($\nu^{\text{opt}} = 3$) | | |
|---|---|---|---|---|---|---|---|---|---|
| SNR | 0.5 | 1.5 | 3 | 0.5 | 1.5 | 3 | 0.5 | 1.5 | 3 |
| $I_n(\omega_j)$ | 0.97 | 0.9735 | 0.9735 | 0.505 | 0.5075 | 0.5045 | 0.9615 | 0.9645 | 0.964 |
| PL(1) | 0.9295 | 0.9655 | 0.977 | 0.9275 | 0.9615 | 0.967 | 0.9615 | 0.9795 | 0.9875 |
| PL($\nu$), $\nu = 1 \ldots \nu^{\text{opt}}$ | 0.94 | 0.981 | 0.991 | 0.928 | 0.962 | 0.9685 | 0.9615 | 0.9795 | 0.988 |
| PL($\nu$), $\nu = 1 \ldots 40$ | 0.94 | 0.981 | 0.991 | 0.928 | 0.962 | 0.9685 | 0.9615 | 0.9795 | 0.988 |

## 7. Clustering Temperature Time Series

We clustered long streams of temperatures, measured by sensors in various weather stations in the US. We considered time series of maximum daily temperatures (TMAX) from 11 December 2016 to 30 September 2019 ($T = 1024$). The length $T$ is a power of two, which was ideal for estimating the second-order spectrum using a Fast Fourier Transform (FFT). In addition, the time span included days from all four seasons in the US. Data for these locations were obtained from the National Oceanic and Atmospheric Administration (www.ncdc.noaa.gov (accessed on 1 October 2019)), and were preprocessed by (i) excluding the series if there were more than 100 missing values, and (ii) imputing a smaller number of missing values in the series, using the function `na_se adec()` in the R package **imputeTS**. We then fitted a linear trend regression model to the time series in $N = 63$ locations, in order to remove the long-term trend.

Figure 1 shows the raw preprocessed temperature series (black) and the detrended series (blue) at four randomly selected locations. Each of the detrended time series at different locations showed a strong annual cycle. The levels and ranges of temperatures varied between locations: for instance, in Albany, the time series exhibited amplitudes ranging from 20 to 100 degrees Celsius, while Albert Lea had amplitudes ranging from $-20$ to around 80 degrees Celsius. The locations also exhibited different patterns of temporal dependence.

We implemented Algorithm 1, to select persistence landscapes of optimal order as features, and then used the K-means algorithm to cluster the $N = 63$ temperature time

series based on these features. The red circle in each plot in Figure 2 denotes the selected order $\nu^{\text{opt}}$ from the elbow method. The method converged quickly (at iteration 5) and chose $\nu^{\text{opt}} = 3$. Note that, because $\nu^{\text{opt}} = 3$ held for two successive iterations, this value was picked as the optimum order for these data. We constructed the persistence landscapes of orders $\nu = 1, 2, 3$, and used these as features in the K-means algorithm. We showed the computing times on a MacBook Pro (16-inch, 2019 with a 2.3 GHz 8-Core Intel Core i9). The code for the optimum $\nu^{\text{opt}} = 3$ landscapes was 27.92 seconds versus 6.04 minutes to run $\nu = 40$ landscapes: that is, using the optimum landscapes produced a speed-up of approximately 13 times.

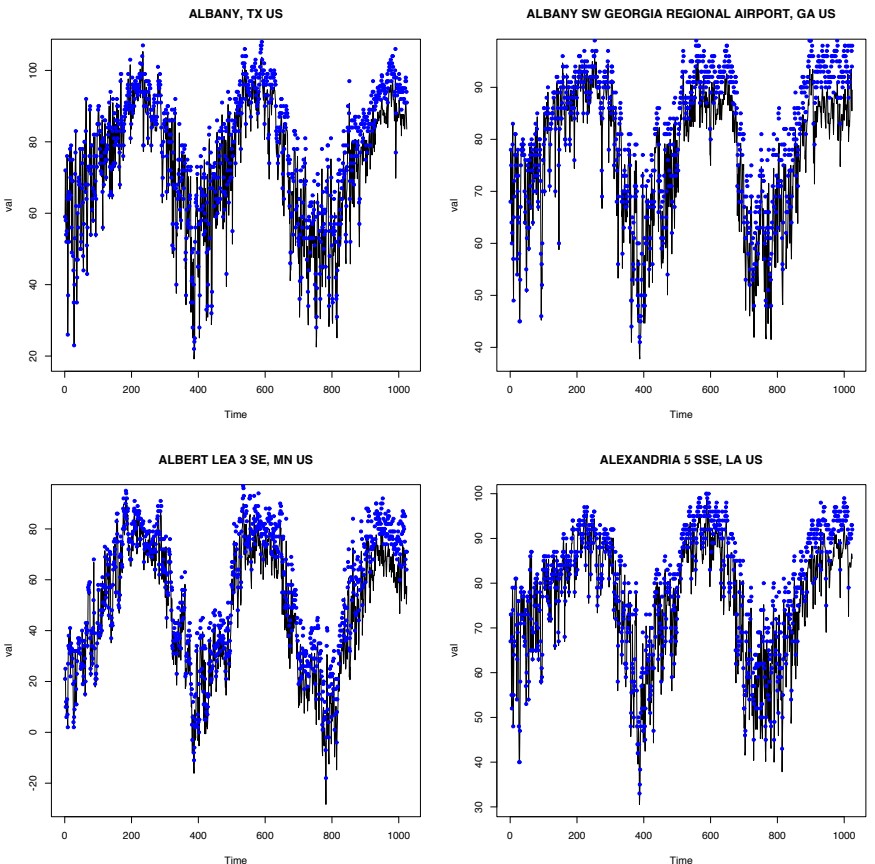

**Figure 1.** Raw temperatures (blue) and detrended temperatures (black) from four locations.

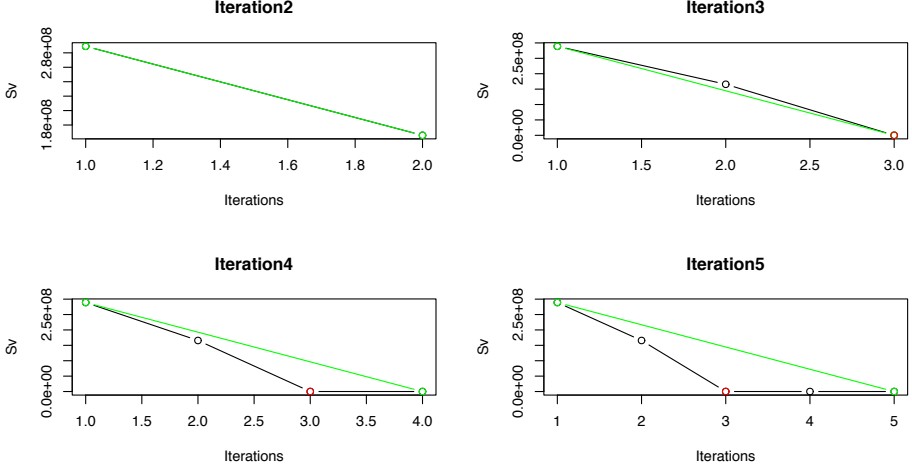

**Figure 2.** Persistence landscape (PL) order selection for temperature time series using the elbow method.

The number of clusters in the K-means method was chosen to be $C = 3$, using the average silhouette method [33] (see Figure 3). The profiles of the daily temperatures within the three clusters are shown in Figure 4. The titles of each plot show the number of time series in each cluster, i.e., $C_1 = 21$, $C_2 = 9$, and $C_3 = 33$. In each plot, the black lines are the time-wise median of the time series, while the blue dots denote the maximum and minimum over time within the cluster. The temperatures in the three clusters clearly exhibit different temporal patterns. For instance, the locations in Cluster 3 have higher temperature values in general than the locations in the other clusters, while Cluster 2 includes locations with a wider range of temperatures.

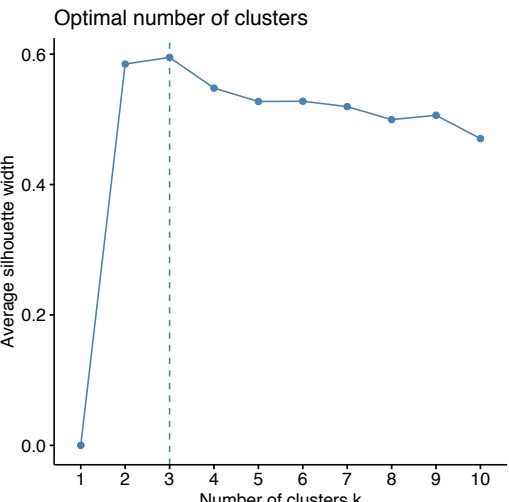

**Figure 3.** Selecting the number of clusters, using the average silhouette method.

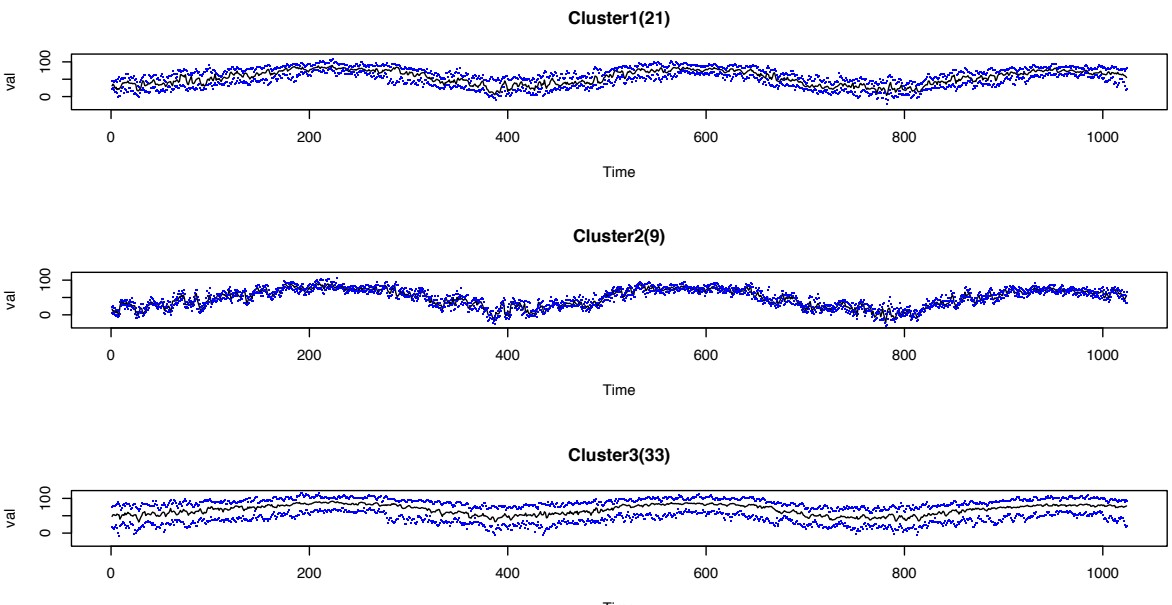

**Figure 4.** Profiles of daily temperatures in the $C = 3$ clusters selected using the K-means method with $\nu^{\mathrm{opt}}$ persistence landscapes as features.

It is useful to visualize the weather stations geographically, as shown in Figure 5, in order to see how the locations in different regions of the US were clustered by our algorithm. The locations in Cluster 2 (the green dots) are dense around the north-central parts of the US, while both Cluster 1 and Cluster 3 appear to be more spread out around the country.

For the most part, the locations in Cluster 1 appear to span the northern part of the country, while the locations in Cluster 3 are at lower latitudes, and are closer to the coast.

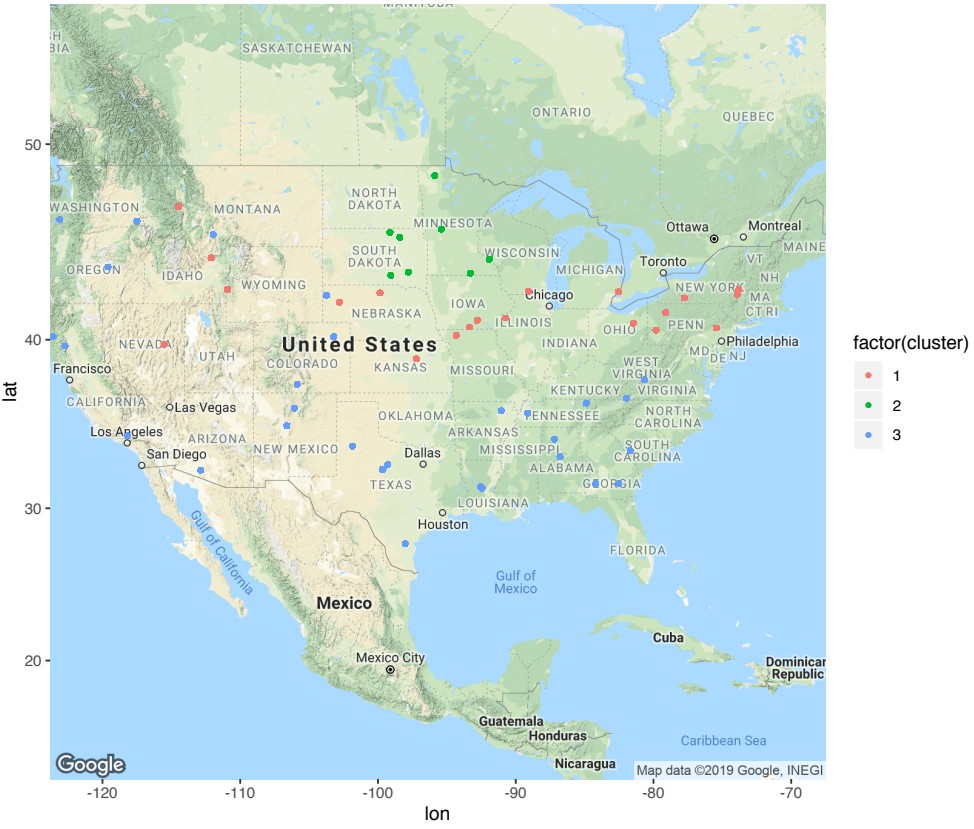

**Figure 5.** Distribution of locations in the $C = 3$ clusters.

It is also interesting to discuss the clustering based on the eight climates in the US (https://scijinks.gov/climate-zones/ (accessed on 2 October 2019)): these include subtropical evergreen broad-leaved forest, subtropical desert and grassland, temperate deciduous broad-leaved forest, temperate grassland, plateau mountain, temperate desert, Mediterranean, temperate maritime, and sub-cold coniferous forest. Most of the locations in Cluster 3 are located in warm climate zones, such as the subtropical evergreen broad-leaved forest zone. The locations in Cluster 1 are mainly located in the mid-south (the temperate deciduous broad-leaved forest zone, and the middle part of the temperate grassland zone). Cluster 2 is mainly located in the northern part of the temperate deciduous broad-leaved forest zone and the temperate grassland zone.

## 8. Conclusions

The use of persistent homology (an important topic in topological data analysis) for analyzing time series is a novel emerging area of scientific enquiry, which connects mathematics, computer science, and statistics. Persistent landscapes are especially useful constructs for clustering a large number of time series with different temporal dependence properties. As lower-order persistence landscapes contain more important topological features than those of higher orders, in many situations it may be unnecessary and computing-intensive to use persistence landscapes of all orders to elicit useful features of time series. Previous research on this topic has not considered the problem of *selecting the order* of the persistence landscapes for effective clustering. In this paper, we present the elbow method, a data-based approach to selecting the optimal order of persistence landscapes. The novelty of our approach was to only use persistence landscapes up to the selected order as features, for

faster clustering of a large number of time series. Note that if we have labeled time series, these features can also be used in supervised learning algorithms (classification).

We evaluated our approach through a simulation study, which showed that using features up to the optimal selected order produced the best clustering performance in most cases. We also applied our algorithm to clustering weather stations in the US, based on daily temperatures from sensors, in interesting and meaningful ways.

It is interesting to note that, unlike the simulated cases, the same clustering labels that we obtained from our algorithm were also obtained for the temperature series by using only the first-order persistence landscape, or all orders of persistence landscapes as features. In future work, it will be useful to demonstrate the usefulness of our algorithm for IoT time series with different types of complex dependence patterns. If the IoT series exhibit nonstationary behavior, we could construct persistent landscapes starting from higher-order spectra, whose fast computation is discussed in [34]. Other useful extensions of our approach would include an investigation of the robustness properties of persistent-homology-based statistical learning for time series, and developing alternatives to the elbow method.

**Author Contributions:** Conceptualization, R.C. and N.R.; methodology, R.C. and N.R.; writing—review & editing, R.C. and N.R. All authors have read and agreed to the published version of the manuscript.

**Funding:** This research received no external funding.

**Data Availability Statement:** Not applicable.

**Conflicts of Interest:** The authors declare no conflict of interest.

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
