# Peer review of "Feature Construction Using Persistence Landscapes for Clustering Noisy IoT Time Series"

_futureinternet, doi:10.3390/fi15060195_

Round 1

Reviewer 1 Report

The manuscript "Feature Construction Using Persistence Landscapes For Clustering Noisy IoT Time Series" proposes a solution to the important engineering problem of efficiently reducing the dimensionality and extracting key features of stochastic time series data generated by IoT devices. The main methodology proposed by the authors highlights an algorithm for selecting the optimal order of persistent landscapes with further clustering of extracted useful attributes of stationary time series based on the selection of the number of clusters by the elbow method. Overall impressions of familiarity with the study are contradictory. On first reading, this manuscript does not arouse reader interest. However, the manuscript is well structured and has some contribution to the development of the study field. The main suggestions for improving the manuscript are general improvements in the style of presentation. Specific issues and comments on the manuscript:

1. The abstract does not offer specific research findings obtained.

2. It is recommended to add a graphical abstract to attract the interest of readers and improve the perception of the study material. Instead of a graphical abstract, a Figure with study stage pipelines may be suitable.

3. In the Introduction section, the problem is revealed unevenly and incompletely. There is a lack of coherence in the material presented. Row 79-93 - needs improvement. It is necessary to formulate, motivation, purpose and tasks of the study more cleanly and clearly.

4. Elements of scientific novelty as the authors' main contribution to the development of scientific topics need to be clearly formulated.

5. The authors declare the "reduction of computational costs while maintaining accuracy". Perhaps it is necessary to give comparisons with other solutions using these benchmarks.

6. A Discussion section should be added to assess the main limitations of the study. Disclose the scope and comprehensiveness of the proposed solution. Reveal the potential of applying this methodology to other IoT devices.

7. Conclusions are primitive and do not reflect the actual research experience. Perhaps Figure 5 should be moved to another section of the manuscript.

Minor editing of English language required

Reviewer 2 Report

-The paper presents a feature-based clustering approach using topological data analysis to cluster noisy IoT time series.

-Typically, citations to references are not placed at the start of a sentence (e.g. "[4] discussed the role of wireless..."). I suggest adding text before the cited references (e.g. "The work presented in [4] discussed the role of wireless...").

-Links to websites are typically placed as footnotes to improve the readability of the document.

-The technical aspects of the research presented seem sound and detailed.

-Regarding the writing style and typos in the paper, I would suggest that the authors to double check the text as there are some typos in the paper.

-Although you presented some previous related work in the "Introduction" section, the actual "Related Work" section only has four previous related works. As a "Related Work" sections is very brief and I think that the number of previous related works is not sufficient.

-Did you consider using another clustering algorithm to compare the results with the ones obtained with K-Means?

-It is not clear the significance and relevance of the clustering temperature time series results and map plot. How can the results presented in the map be used or useful in the real world?

-You should have explained the rationale of using the range dates considered for the time series of maximum daily temperatures.

-In Figure 2, the text above the four plots seems to read "Iteratlon", instead of "Iteration".

-The paper presents a feature-based clustering approach using topological data analysis to cluster noisy IoT time series.

-Typically, citations to references are not placed at the start of a sentence (e.g. "[4] discussed the role of wireless..."). I suggest adding text before the cited references (e.g. "The work presented in [4] discussed the role of wireless...").

-Links to websites are typically placed as footnotes to improve the readability of the document.

-The technical aspects of the research presented seem sound and detailed.

-Regarding the writing style and typos in the paper, I would suggest that the authors to double check the text as there are some typos in the paper.

-Although you presented some previous related work in the "Introduction" section, the actual "Related Work" section only has four previous related works. As a "Related Work" sections is very brief and I think that the number of previous related works is not sufficient.

-Did you consider using another clustering algorithm to compare the results with the ones obtained with K-Means?

-It is not clear the significance and relevance of the clustering temperature time series results and map plot. How can the results presented in the map be used or useful in the real world?

-You should have explained the rationale of using the range dates considered for the time series of maximum daily temperatures.

-In Figure 2, the text above the four plots seems to read "Iteratlon", instead of "Iteration".

Round 2

Reviewer 1 Report

It`s well done. Manuscript has been recomindation for publication.

It`s done

Reviewer 2 Report

I think the authors have addressed my comments and the comments of the other reviewer appropriately and in the best way they could. I think the manuscript has been sufficiently improved to warrant publication.